# Transformer Compression via Subspace Projection

## Abstract

We propose TCSP, a novel method for compressing a transformer model by focusing on reducing the hidden size of the model. By projecting the whole transform model into a subspace, we enable matrix operations between the weight matrices in the model and features in a reduced-dimensional space, leading to significant reductions in model parameters and computing resources. To establish this subspace, we decompose the feature matrix, derived from different layers of sampled data instances, into a projection matrix. For evaluation, TCSP is applied to compress T5 and BERT models on the GLUE and SQuAD benchmarks. Experimental results demonstrate that TCSP achieves a compression ratio of 44% with at most 1.6% degradation in accuracy, surpassing or matching prior compression methods. Furthermore, TCSP exhibits compatibility with other methods targeting filter and attention head size compression.

## 1    Introduction

The transformer model [1] is widely used in Natural Language Processing as well as other domains such as Computer Vision [2, 3, 4] and Speech Recognition [5, 6, 7]. Despite its impressive performance, the large size of transformer models and the high inference latency limit their practical deployment. To address this challenge, model compression techniques including pruning [8, 9, 10, 11, 12, 13] and low-rank decomposition [14, 15, 16, 17] have been proposed to reduce model parameters and improve inference speed while ensuring that the performance of the compressed model is not significantly disturbed.

The transformer model consists of a series of weight matrices that are determined by the hidden size $d$, attention head size $d_h$ in the multi-head attention layers, and the number of filters $d_f$ in the feed-forward network layers. Existing compression methods primarily focus on reducing the attention head size $d_h$ [14, 8, 13] and the number of filters $d_f$ [8, 13], or performing matrix decomposition to transform $d \times d$ matrices into two $d \times k$ matrices [14, 15, 16]. However, none of these methods directly address the reduction of the hidden size $d$. Although CoFi [13], a pruning method, attempts to compress $d$, it can only achieve a 1% reduction while keeping model performance.

This paper introduces a novel method, named **T**ransformer **C**ompression via **S**ubspace **P**rojection (TCSP), for compressing transform models by reducing the hidden size. By projecting the transformer model into a subspace, TCSP achieves this compression. To create this subspace, we employ low-rank factorization to decompose the feature matrix derived from sampled data instances into a projection matrix. Specifically, we gather multiple data instances, pass each through the transformer model to obtain their respective features from different layers, and concatenate them to form a feature matrix. Decomposing this matrix yields the projection matrix, which allows us to project the model into a subspace, resulting in a significant reduction in model parameters. Furthermore, TCSP can be easily combined with other compression methods that reduce attention head size and the number of filters, ensuring compatibility. Although low-rank factorization is employed in TCSP, it is considered a sub-technique to achieve the primary goal of reducing hidden size.

Submitted to 37th Conference on Neural Information Processing Systems (NeurIPS 2023). Do not distribute.

Our contributions can be summarized as follows:

- We take a fresh perspective on model compression by reducing the hidden size of the transformer model, which has been rarely explored before.

- We propose the TCSP technique to achieve the compression goal. TCSP decomposes the feature matrix derived from sampled data instances into a projection matrix, which is then used to project the transformer model into a subspace. In addition, TCSP is compatible with other compression methods, such as reducing the multi-head attention size and the number of filters, which could further speed up inference.

- Experimental results on two widely-used benchmarks, GLUE and SQuAD, show that TCSP can compress 44% of the parameters of both T5 and BERT within 1.6% accuracy loss, surpassing or matching prior compression methods.

## 2    Related Work

The transformer model is a general model that has been widely used in the fields of deep learning. To improve inference speed and reduce memory overhead, different approaches have been proposed to compress the transformer. Previous research [18] categorize these approaches into five distinct categories: quantization [19, 20, 21], pruning [8, 9, 10, 11, 12, 13], knowledge distillation [22, 23], low-rank factorization [14, 15, 16, 17], and weight sharing [24, 25]. These five types of methods are orthogonal to each other. In this work, we focus on exploring low-rank factorization and pruning approaches as means to directly reduce the number of parameters in fined-tuned task-specific models. We do not delve into Knowledge distillation and weight sharing, as they involve training models from scratch. We do not cover quantization, which compresses models based on storage considerations.

**Low-Rank Factorization for Transformer.**  In recent years, various low-rank decomposition methods have been proposed specifically for Transformers. Ma et al. [26] introduce a block term tensor decomposition approach to decompose the multi-head attention in Transformers. Noach et al. [27] propose a two-stage method for Transformer compression, where they first employ SVD to decompose Transformer's weight matrix and then fine-tune the model using knowledge distillation. However, direct SVD-based compression of the weight matrix assumes that it possesses low-rank properties, which may not always hold for Transformers. To address this issue, Chen et al. [14] propose a data-aware low-rank factorization method. This method minimizes the reconstruction error of the matrix multiplication between each weight matrix and its corresponding input feature matrix rather than directly minimizing the reconstruction error of the weight matrix itself. Hsu et al. [15] and Hua et al. [16] propose FWSVD and TFWSVD, respectively, which utilize Fisher information to measure the contribution of different parts of the weight matrix to model performance during the SVD process, achieving improved results compared to direct SVD. Alternatively, Tahaei et al. and Edalati et al. [28, 29] employ Kronecker decomposition to preserve the matrix rank and successfully compress models like BERT and GPT2.

**Pruning for Transformer**.  Pruning is a commonly used technique for eliminating unnecessary parameters in the model. Existing pruning methods can be broadly divided into two categories: unstructured pruning and structured pruning. Unstructured pruning aims to remove unimportant scalar values in model's parameters. Various unstructured pruning approaches have been proposed for Transformer, such as magnitude-based [30], first-order [31], second-order [32], and lottery ticket hypothesis [33]. Although unstructured pruning algorithms can remove many redundant parameters while ensuring accuracy, compressed models require specific sparse data structures and hardware support to take advantage of unstructured pruning. For this reason, structure pruning approaches [8, 9, 10, 11, 12, 13] are proposed to remove weight blocks in the transformer model, including entire transformer layers, attention heads, and filters. Unlike unstructured pruning, structure pruning can accelerate inference speed and reduce memory overhead without specialized data structure and hardware.

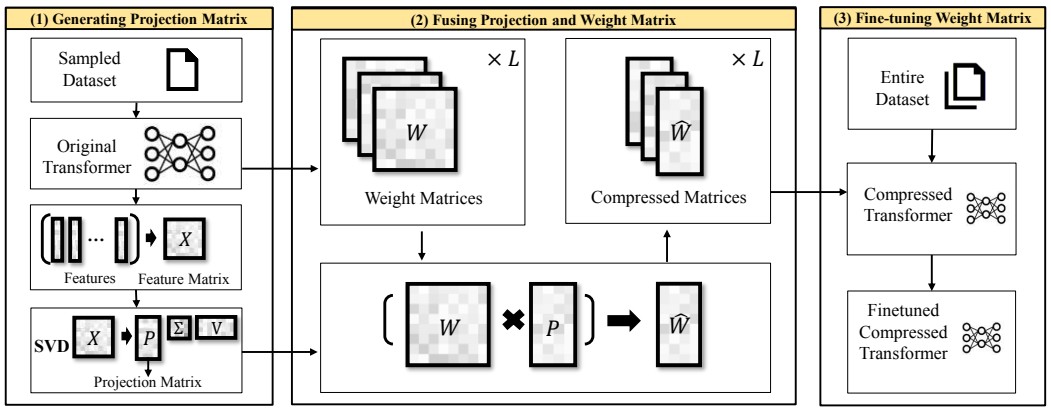

Figure 1: Workflow of Transformer Compression via Subspace Projection (TCSP).

## 3 Overview

### 3.1 Preliminary: Transformer Architecture

We take T5 [34] as an example of the transformer model to present the proposed TCSP, although it can be easily extended to other transform models such as BERT [35] and GPT [36].

A transformer model comprises a stack of blocks, each containing a multi-head attention (MHA) layer, an optional cross-attentive (CA) layer, and a feed-forward network (FFN) layer.

**Multi Head Attention (MHA).** In a transformer encoder, each MHA layer consists of $H$ attention heads that facilitate interactions between tokens, with a pre-normalization step. Other transform models such as BERT adopt post-normalization after the MHA layer.

$$x_{\mathrm{M}} = x + \mathrm{MHA}(x_n), \quad x_n = \mathrm{norm}(x), \quad \mathrm{MHA}(x_n) = \sum_{i=1}^{H} \mathrm{Att}^{(i)}(x_n), \tag{1}$$

$$\mathrm{Att}^{(i)}(x_n) = W_O^{(i)\top} W_V^{(i)} x_n \cdot \mathrm{Softmax}((W_K^{(i)} x_n)^\top (W_Q^{(i)} x_n)/\sqrt{d}), \tag{2}$$

where $\mathrm{norm}(x)$ represents the normalization function such as LayerNorm and RMSNorm, $W_Q, W_K, W_V, W_O \in \mathbb{R}^{d_h \times d}$ are the parameters of an MHA layer, denoting the query, key, value, and output matrices, respectively. Here $d$ denotes the hidden size, and $d_h = d/H$ denotes the attention head size.

In a transformer decoder, a CA layer is added after the MHA layer. The CA layer closely resembles the MHA, with one only distinction: the feature matrices multiplied by $W_Q$ and $W_K$ in the CA layer are sourced from the output of the Transformer encoder. Further information on the CA layer can be found in Appendix A.

**Feed Forward Network (FFN).** Following each MHA layer, there is an FFN layer that takes $x_M$ as input and generates $x_F$ as output:

$$x_F = x_M + \mathrm{FFN}(x_n), \quad x_n = \mathrm{norm}(x_M), \quad \mathrm{FFN}(x_n) = W_D^\top \sigma(W_U x_n), \tag{3}$$

where $\sigma$ is the activation function, and $W_U, W_D \in \mathbb{R}^{d_f \times d}$ are the parameters of the FFN layer, which correspond to the up and down matrices, respectively. Here $d_f$ indicates the number of filters.

### 3.2 Basic Idea of TCSP: Subspace Projection

The basic idea of **T**ransformer **C**ompression via **S**ubspace **P**rojection (TCSP) is to project the Transformer model into a suitable subspace via matrix fusion. To provide a clearer explanation, we take the linear regression model as an example. Here we denote the original model $\mathcal{F}$ as:

$$\mathcal{F}(x) = Wx, \tag{4}$$

where $W \in \mathbb{R}^{1 \times d}, x \in \mathbb{R}^d$.

Assume that the input $x$ of the model is distributed in a $k$-dimensional subspace. In other words, there exists a projection matrix $P \in \mathbb{R}^{d \times k}$ such that every $x$ in the dataset satisfies $x = P\hat{x}$, where $\hat{x} \in \mathbb{R}^k$. Thus, we have,

$$\hat{\mathcal{F}}(\hat{x}) = (WP)\hat{x} = \hat{W}\hat{x}, \tag{5}$$

where $\hat{W} = WP \in \mathbb{R}^{1 \times k}$ represents the compressed weight matrix, $\hat{x} \in \mathbb{R}^k$. In this way, we project the original model from a $d$-dimensional space into a reduced $k$-dimensional subspace. We will elaborate on how to extend this method to the transformer model.

**Workflow of TCSP.** The proposed TCSP comprises three stages, as illustrated in Figure 1. Firstly, given the training data and the original transformer model (e.g., a fine-tuned T5), we sample a subset of the training data, feed it into the transformer model to obtain the feature matrix $X$, and employ SVD on $X$ to derive the projection matrix $P$. Then, we project the weight matrices $\{W\}$ of the original transformer to a subspace via the fusion of the model's weight matrices and the projection matrix, resulting in $\{\hat{W}\}$. Finally, following prior work [14, 15, 16], we fine-tune the compressed transformer with the entire training data.

# 4 Methodology

## 4.1 Generating Projection Matrix

Motivated by the fact that features produced by the transformer model usually tend to reside in a low-dimensional subspace, our objective is to carry out the forward pass of the transformer model in the low-dimensional subspace. To achieve this, the initial step of TCSP focuses on determining the subspace where the features are located. This can be formalized as a optimization problem:

$$\underset{P}{\arg\min} \; \underset{x \in \mathbb{D}}{\mathbb{E}} \left( \sum_{i=1}^{N} \|L^{(1 \sim i)}(x) - PP^\top L^{(1 \sim i)}(x)\|_F^2 \right) \quad \text{s.t.} \quad P^\top P = \mathbf{I}, \tag{6}$$

where $N$ denotes the number of layers in the transformer, $L^{(i)}$ represents the $i$-th layer in the transformer model, which can correspond to different components such as the MHA or the FFN layer. $L^{(1 \sim i)} = L^{(i)} \circ \ldots L^{(2)} \circ L^{(1)}$ represents the composite function that is formed by sequentially applying the transformations from the first layer to the $i$-th layer. $L^{(1 \sim i)}(x)$ refers to the application of the composition function $L^{(1 \sim i)}$ on the input $x$, resulting in its corresponding feature. $\mathbf{I}$ is an identity matrix, and $P$ is the desired projection matrix. For a given projection matrix $P$, $P^\top x$ can be interpreted as projecting $x$ from the original space into the subspace corresponding to the matrix $P$, while $PP^\top x$ can be viewed as projecting $P^\top x$ from the subspace back into the original space. Therefore, we can use $x - PP^\top x$ to measure whether the vector $x$ belongs to the subspace.

Eq. 6 is equivalent to the following equation, the proof of which is shown in Appendix B.

$$\underset{P}{\arg\min} \|X - PP^\top X\|_F^2 \quad \text{s.t.} \quad P^\top P = \mathbf{I}, \tag{7}$$

where

$$X = \left[ L^{(1)}(x_1), \cdots, L^{(1)}(x_M), L^{(1 \sim 2)}(x_1), \cdots, L^{(1 \sim 2)}(x_M), L^{(1 \sim N)}(x_1), \cdots, L^{(1 \sim N)}(x_M) \right]. \tag{8}$$

Let $l_m$ denote the sequence length of the $m$-th data instance, and $d$ denote the hidden size. Consequently, the shape of $X$ can be expressed as $d \times (N \times \sum_{m=1}^{M} l_m)$, where $N$ is the number of layers, and $M$ is the number of the sampled data instances.

Such an optimization problem in Eq. 7 can be well solved by SVD [37].

$$U, \Sigma, V^\top = \text{SVD}(X), \tag{9}$$

where $U$ and $V$ are two orthogonal matrices, and $\Sigma$ is a diagonal matrix.

Then, the first $k$ columns of the matrix $U$ (i.e., the top-$k$ important eigenvectors of the feature matrix $X$) compose the desired projection matrix $P$.

$$P = U_{:,:k} \tag{10}$$

Appendix G.1 describes the process of generating the projection matrix $P$. Notably, although we sample a subset of data instances for computing feature matrix $X$, the size of $X$ is still large, resulting in a significant overhead of SVD. To mitigate this, we only select a subset of columns from $X$ to create a submatrix for SVD. Following previous work [14, 8], we approximate the SVD by sampling 2,000 tokens, which corresponds to the number of columns in matrix $X$. In the experiments, we examine the impact of the number of sampled tokens on the performance of TCSP.

## 4.2 Fusing Projection and Weight Matrix

Once the projection matrix $P$ for the the transformer model has been obtained, we can construct an approximation of the original model within the subspace using the projection matrix. The basic idea behind this is to fuse the projection matrix with the weight matrices of the transformer.

Given an input $x$, we pass it through the transformer model starting from the first layer to the last $N$-th layer, resulting in the corresponding feature vector $L^{(1\sim N)}(x)$. According to Eq.6, we can use the projection matrix $P$ to add dimensionality reduction and dimensionality enhancement operations between each layer, while preserving the final outcome, as shown in the following equation,

$$L^{(1\sim N)}(x) = L^{(N)} \circ \cdots \circ L^{(2)} \circ L^{(1)} \tag{11}$$

$$\approx U \circ D \circ L^{(N)} \circ \cdots \circ L^{(2)} \circ U \circ D \circ L^{(1)} \circ U \circ D(x) \tag{12}$$

$$\approx U \circ \hat{L}^{(N)} \circ \cdots \circ \hat{L}^{(2)} \circ \hat{L}^{(1)} \circ D(x) \tag{13}$$

$$\approx P \hat{L}^{(1\sim N)}(P^\top x), \tag{14}$$

where $U(x) = Px$ represents the dimensionality enhancement operation, $D(x) = P^\top x$ represents the dimensionality reduction operation, $\hat{L}_i = D \circ L_i \circ U$ is the projected layer, and $\hat{L}^{(1\sim N)} = \hat{L}^{(N)} \circ \cdots \circ \hat{L}^{(2)} \circ \hat{L}^{(1)}$ is the projected model.

According to Section 3.1, any original layer $L^{(i)}(x)$ can be formulated as:

$$L(x) = x + \text{Layer}(\text{norm}(x)), \tag{15}$$

where the function "Layer" can be instantiated by various components such as MHA, CA, and FFN. The projected layer $\hat{L}$ operates on a $k$-dimensional feature $\hat{x}$ as input and produces a corresponding $k$-dimensional feature $\hat{L}(\hat{x})$, i.e.

$$\hat{L}(\hat{x}) = P^\top(P\hat{x} + \text{Layer}(\text{norm}(P\hat{x})) = \hat{x} + P^\top \text{Layer}(\text{norm}(P\hat{x}). \tag{16}$$

Compared with the $d$-dimensional input and output vectors of the original layer $L(x)$, the projected layer $\hat{L}(\hat{x})$ in Eq.16 reduces the dimensions of the layer's input and output to $k$. However, this reduction in dimensionality does not alleviate the computational effort. Therefore, we propose two methods to compress the parameters within the projected layer: (1) Matrix fusion, which merges the multiple consecutive matrix multiplications into a single matrix. (2) Normalization function reconstruction, which is used to swap the computational order of matrix operations and normalized functions to increase the chance of matrix fusion. Next, we first explain the normalization function

reconstruction and then discuss how the matrix fusion technique can be employed to compress the MHA and FFA layers.

**Normalization Function Reconstruction**. Within the projected layer $\hat{L}$ as shown in Eq.16, we encounter the computation of $\mathrm{norm}(P\hat{x})$. Additionally, we know that the operation following $\mathrm{norm}(P\hat{x})$ is a matrix multiplication. Thus, if we can find a new matrix $\hat{P}$ along with a normalization function $\hat{\mathrm{norm}}$ that satisfies $\mathrm{norm}(P\hat{x}) = \hat{P}\hat{\mathrm{norm}}(\hat{x})$, we can compress the parameters of the transformer model by fusing the matrix $\hat{P}$ with the subsequent matrix.

The normalization function in T5 is RMSNorm [38]:

$$\mathrm{norm}(x) = \gamma \frac{x}{\frac{1}{\sqrt{d}}\|x\|}. \tag{17}$$

Therefore, we have

$$\mathrm{norm}(P\hat{x}) = \gamma \frac{P\hat{x}}{\frac{1}{\sqrt{d}}\|P\hat{x}\|} = (\sqrt{\frac{d}{k}}\gamma P)\mathbf{I}\frac{\hat{x}}{\frac{1}{\sqrt{k}}\|\hat{x}\|} = \hat{P}\hat{\mathrm{norm}}(\hat{x}), \tag{18}$$

where $\hat{P} = \sqrt{\frac{d}{k}}\gamma P$, $\hat{\mathrm{norm}}(\hat{x}) = \mathbf{I}\frac{\hat{x}}{\frac{1}{\sqrt{k}}\|\hat{x}\|}$ (note that $\|\hat{x}\| = \|P\hat{x}\|$).

By incorporating Eq.18 into Eq.16, we can define

$$\hat{\mathrm{Layer}}(\hat{x}) = P^{\top}\mathrm{Layer}(\hat{P}\hat{\mathrm{norm}}(\hat{x})) = P^{\top}\mathrm{Layer}(\hat{P}\hat{x}_n), \tag{19}$$

where $\hat{\mathrm{norm}}(\hat{x})$ is abbreviated as as $\hat{x}_n$.

For detailed information on reconstructing other normalization functions such as LayerNorm [39] and BatchNorm [40], please refer to the appendix D. The post-normalization reconstruction is also explained in appendix D.

**MHA Layer Compression**. We can compress the transformer model's parameters by fusing multiple consecutive matrix multiplications into a single matrix. For the MHA layer, we have

$$\hat{\mathrm{MHA}}(\hat{x}_n) = \sum_{i=1}^{H} P^{\top}W_O^{(i)\top}W_V^{(i)}\hat{P}\hat{x}_n \cdot \mathrm{Softmax}((W_K^{(i)}\hat{P}\hat{x}_n)^{\top}(W_Q^{(i)}\hat{P}\hat{x}_n)/\sqrt{d}) \tag{20}$$

$$= \sum_{i=1}^{H} \hat{W}_O^{(i)\top}\hat{W}_V^{(i)}\hat{x}_n \cdot \mathrm{Softmax}((\hat{W}_K^{(i)}\hat{x}_n)^{\top}(\hat{W}_Q^{(i)}\hat{x}_n)/\sqrt{d}), \tag{21}$$

where $\hat{W}_O^{(i)} = W_O^{(i)}P, \hat{W}_V^{(i)} = W_V^{(i)}\hat{P}, \hat{W}_K^{(i)} = W_K^{(i)}\hat{P}, \hat{W}_Q^{(i)} = W_Q^{(i)}\hat{P}$. The shape of these matrices is transformed from $d_h \times d$ to $d_h \times k$, resulting in a reduction of $k/d$ of the original number of parameters. We introduce the CA layer compression in Appendix E.

**FFN Layer Compression**. For the FFN layer, we have

$$\hat{\mathrm{FFN}}(\hat{x}_n) = P^{\top}W_D^{\top}\sigma(W_U\hat{P}\hat{x}_n) = \hat{W}_D^{\top}\sigma(\hat{W}_U\hat{x}_n), \tag{22}$$

where $\hat{W}_D = W_D P, \hat{W}_U = W_U\hat{P}$, which has the same compression rate as the MHA Layer. The complete compression process is shown in Appendix G.2.

## 4.3 Combining TCSP with Other Compression Algorithms

TCSP compresses the hidden size $d$, and we can further compress the number of filters $d_f$ and attention head size $d_h$ by the prior pruning and low-rank factorization methods.

**Filter Pruning**. We can remove the filters in the FFN layer by pruning. Following prior work [13, 8], we introduce mask variables $\text{diag}(m)$ into the FFN layer, where $m$ indicates which filters in the FFN layer are to be retained.

$$\text{FFN}(x; m) = W_D^\top \text{diag}(m) \sigma(W_U x). \tag{23}$$

Then the pruning problem can be formalized as an optimization problem on the mask.

$$\arg\min_m \mathcal{L}(m) \quad s.t. \quad \text{cost}(m) \leq C. \tag{24}$$

Our proposed TCSP method only involves the forward computation process of the model. However, most prior methods find the optimal mask by leveraging the derivative of the loss function with respect to the mask variable $\frac{\partial \mathcal{L}}{\partial m}$. As a result, TCSP cannot be directly integrated into the same framework as previous pruning algorithms. To address this, we propose TCSP-pruning, a filter pruning algorithm that exclusively relies on the forward process of the model, enabling the unifying of the pruning algorithm into the TCSP framework.

TCSP-pruning employs a greedy algorithm to remove the least important filter at each step based on the following score function:

$$\text{score}(i) = |W_{D,i}| * (\text{E}[\sigma(W_{U,i} x)] + \text{std}[\sigma(W_{U,i} x)]), \tag{25}$$

where $\text{score}(i)$ represents the significance of the $i$-th filter. We believe that the importance of the $i$-th filter is influenced by the factors such as the norm of $W_{D,i}$ (the weight matrix associated with the $i$-th filter), the average activation value, and the variance of activation values. The pseudo-code of TCSP-pruning is shown in the Appendix G.3.

**Head Size Compression**. Based on the low-rank property of data distribution, we can further compress the head size of the MHA layer. The objective can be expressed as:

$$\arg\min_M \mathbb{E}_x \|(W_K x)^\top (W_Q x) - (W_K x)^\top M(W_Q x)\|_F^2 \quad s.t. \quad \text{rank}(M) = k, \tag{26}$$

$$\arg\min_M \mathbb{E}_x \|W_O^\top W_V x - W_O^\top M' W_V x\|_F^2 \quad s.t. \quad \text{rank}(M') = k. \tag{27}$$

DRONE [14] gives the solution to the above two optimization problems. Assuming that the optimal solution obtained from the first optimization problem is $M^*$. Afterward, we decompose $M^*$ into the product of two matrices $M^* = U_M^\top V_M$ through SVD. Then, we can compress matrices $W_Q, W_K$ through $U_M$ and $V_M$ ($\hat{W}_K = U_M W_K, \hat{W}_Q = V_M W_Q$). The second problem is addressed in the same way. We can also formalize Eq.7 as $\arg\min_M \|X - MX\|_F^2 \ s.t. \ \text{rank}(M) = k$, but the optimal solution obtained by both formulas is the same.

# 5 Experiment

## 5.1 Experimental Setup

To evaluate our method, we first fine-tune $\text{T5}_{\text{base}}$ and $\text{BERT}_{\text{base}}$ on the training data of each task of the GLUE[41] and SQuAD [42] benchmarks to obtain the baseline models, and then adopt the proposed TCSP or the comparison compression methods to compress these baseline models. On each task, for TCSP, we sample 2000 instances from its training data to produce the project matrix, using it to compress the baseline models and finally fine-tune the compressed models using the entire training data. For the GLUE benchmark, We report accuracy for the MNLI [43], QQP [44], QNLI [41], and SST2 [45] tasks, as well as F1 score and spearman correlation for the MRPC [46] and STSB [41] tasks. For the SQuAD benchmark, we report the F1 score. For more comprehensive information regarding the experimental setup, please refer to Appendix F.

Table 1: Performance of TCSP on $T5_{base}$ and $BERT_{base}$ with various compression rate. "Avg. Diff" refers to the average accuracy difference observed before and after applying model compression on the GLUE and SQuAD datasets. "Speed Up" represents the rate of inference time speedup achieved by "w TCSP{25%, 25%}" compared to the baseline model. "ft." denotes fine-tuning.

| | MNLI | QQP | QNLI | SST-2 | STS-B | MRPC | $SQuAD_{1.1}$ | $SQuAD_{2.0}$ | Avg. Diff |
|---|---|---|---|---|---|---|---|---|---|
| $T5_{base}$ | 86.8 | 91.4 | 93.2 | 94.5 | 90.0 | 91.9 | 88.6 | 79.3 | |
| w TCSP {25%, 0%} | 85.0 | 90.1 | 91.4 | 93.0 | 88.9 | 89.1 | 83.9 | 71.9 | |
| w TCSP {25%, 0%} + ft. | 86.2 | 91.2 | 92.5 | 93.2 | 90.1 | 91.3 | 86.8 | 78.0 | -0.6/-1.3 |
| w TCSP {25%, 25%} | 83.5 | 88.0 | 90.8 | 91.4 | 86.6 | 90.4 | 79.2 | 66.9 | |
| w TCSP {25%, 25%} + ft. | 85.5 | 90.7 | 91.8 | 92.8 | 89.5 | 91.5 | 87.3 | 77.7 | -1.0/-1.5 |
| Speed Up | ×1.25 | ×1.06 | ×1.08 | ×1.36 | ×1.22 | ×1.25 | ×1.15 | ×1.20 | |
| $BERT_{base}$ | 84.4 | 91.1 | 91.4 | 92.2 | 88.4 | 89.9 | 88.5 | 75.8 | |
| w TCSP {25%, 0%} | 31.8 | 68.7 | 49.5 | 49.2 | 68.5 | 1.0 | 13.5 | 11.2 | |
| w TCSP {25%, 0%} + ft. | 83.7 | 91.0 | 91.0 | 92.2 | 88.5 | 90.1 | 86.7 | 76.6 | -0.2/-0.5 |
| w TCSP {25%, 25%} | 31.8 | 67.6 | 49.8 | 49.1 | 67.5 | 0.0 | 12.9 | 10.6 | |
| w TCSP {25%, 25%} + ft. | 83.5 | 90.8 | 90.6 | 91.9 | 87.7 | 89.1 | 87.7 | 76.0 | -0.6/-0.3 |
| Speed Up | ×1.16 | ×1.02 | ×1.62 | ×2.36 | ×1.96 | ×3.93 | ×1.36 | ×1.36 | |

Table 2: Performance comparison with prior compression methods using $BERT_{base}$ as the baseline.

| | Sajjad et al.(33.3%) | Kwon et al.(40%) | DRONE(-) | FWSVD(40%) | TFWSVD(40%) | TCSP(40%) |
|---|---|---|---|---|---|---|
| QQP | 90.6(-0.5) | 90.4(-0.6) | 90.1(-0.8) | $87.6^*(-0.2)$ | $86.9^*(-0.9)$ | 90.8(**-0.3**) |
| QNLI | 89.7(-1.4) | 90.0(-1.4) | 89.3(-2.1) | 89.5(-1.8) | 90.3(-1.0) | 90.6(**-0.8**) |
| SST-2 | 90.6(-1.8) | 92.5(**-1.1**) | 90.8(-1.5) | 91.2(-1.8)) | 91.1(-1.9) | 91.1(**-1.1**) |
| MRPC | 79.4(-8.6) | 85.3(-1.0) | 88.0(-1.5) | 88.0(+0.6) | 89.0(**+1.6**) | 89.1(-0.8) |

## 5.2 Performance Evaluation

Table 1 shows the accuracy results of $T5_{base}$ and $BERT_{base}$ with different compression rates. The proposed TCSP can reduce the hidden size $d$. Additionally, in Section 4.3, we introduce a filter pruning method that can reduce the number of filers $d_f$. Moreover, the existing DRONE [14] can compress the attention head size $d_h$. In TCSP{a%, b%}, the notation a% denotes the compress rate of hidden size, while b% denotes the compress rate of the attention head size plus the number of filters. Consequently, the overall compression rate of the model is a% + b% - a% * b%.

When employing TCSP with a compression rate of {25%, 0%} which retains 75% hidden size while preserving the original attention head size and number of filters, we observe that the compressed $T5_{base}$ and $BERT_{base}$ models exhibit a maximum drop in accuracy of only 1.3% on the GLUE and SQuAD datasets. Building upon this, We further apply filter pruning and compress the head size, resulting in a model denoted as TCSP {25%, 25%}. Remarkably, this additional compression does not negatively impact the baseline performance, highlighting the compatibility of TCSP with the filter pruning and attention head size compression methods. Overall, we achieve a compression rate of 44% for both $T5_{base}$ and $BERT_{base}$ models with only a 1.6% loss in accuracy. Like prior work [14, 15, 16], fine-tuning is necessary for ensuring performance. For more detailed information on model performance at various compression rates, please refer to Appendix F.5.

## 5.3 Comparison with Prior Methods

We conduct a comprehensive comparison of TCSP with prior methods, considering both their performance and compression time cost.

Following the setting adopted by Kwon et al. [8], we compare TCSP with prior pruning methods and low-rank factorization methods. The evaluation involves compressing models with compression rate constraints on four tasks within the GLUE benchmark: QQP, QNLI, SST-2, and MRPC. Notably, we conduct this evaluation without employing knowledge distillation. The comparison methods include Sajjad et al. [47], Kwon et al. [8], DRONE [14], FWSVD [15], and TFWSVD [16]. To ensure a fair comparison, we select $BERT_{base}$ as the baseline model, as it is commonly employed across all the comparison methods. To assess the impact of compression, we compare the amount of accuracy drop

Table 3: Compression time cost comparison with prior compression methods using $BERT_{base}$ as the baseline evaluated on the MNLI dataset. "# Epochs" represents the number of epochs to fien-tune the model weights on the entire training data.

|  | DynaBERT [23] | EBERT [11] | BMP [12] | CoFi [13] | Kwon et al. [8] | TCSP |
|---|---|---|---|---|---|---|
| Time cost (hr) | 12 | 5 | 17 | 33 | 0.01 | 2.16 (0.16 + 2.00) |
| # Epochs | 4 | 6 | 20 | 40 | 0 | 2 |

Table 4: Effect of sampled tokens.

| Number of tokens | QQP | QNLI | SST-2 | MRPC |
|---|---|---|---|---|
| 1K | 91.2 | 92.3 | 93.8 | 91.5 |
| 2K | 91.2 | 92.5 | 93.2 | 91.3 |
| 4K | 91.2 | 92.5 | 93.7 | 91.8 |

Table 5: Effect of SVD

| Method | QQP | QNLI | SST-2 | MRPC |
|---|---|---|---|---|
| TCSP-SVD | 91.2 | 92.5 | 93.2 | 91.3 |
| TCSP-Random | 63.1 | 49.4 | 49.0 | 0.0 |

experienced by each method since the absolute accuracy of the baseline $BERT_{base}$ may vary slightly across different papers. The compression rate is indicated in parentheses after each method's name, except for DRONE, as their paper does not explicitly report the compression rate.

Table 2 presents the performance of the compressed models and the difference from their corresponding baselines for all the compression methods under comparison. The performance of FWSVD and TFWSVD is marked with an asterisk since they use the F1 metric instead of accuracy specifically on the QQP dataset. However, this metric does not impact the comparison of the metric difference. Our method TCSP demonstrates comparable or superior results compared to the prior methods (A lower accuracy drop indicates better performance).

To assess the time cost, we evaluate the performance on the MNLI dataset, which is larger than other datasets in GLUE. Table 3 shows that TCSP requires only 0.16 hours for model compression and an additional 2 hours to model fine-tuning. This indicates that TCSP is significantly faster than most of the comparison methods, with the exception of the pruning method proposed in Kwon et al. [8]. However, the pruning method can be effectively combined with TCSP, allowing for compatibility and further optimization.

## 5.4 Ablation Study

To prevent the computation of SVD on excessively large matrices, We sample a subset of columns from the feature matrix. In other words, we randomly choose several tokens within each instance to compute the projection matrix. Consequently, we evaluate the effect of the sampled tokens on the compression performance. Table 4 presents the results of this analysis, revealing that satisfactory outcomes can be achieved when the number of tokens (1,000) is greater than the hidden size of the model (768).

Furthermore, we explore the influence of SVD. In Table 5, we replace the projection matrix generated by SVD with a randomly generated matrix and show the experimental results in Table 5. It is observed that using a random matrix for compression significantly diminishes the model's performance. Therefore, it is necessary to use SVD to compute the projection matrix.

## 6 Conclusion

This paper proposes TCSP, a method for compressing the transformer model by leveraging subspace projection. TCSP employs SVD on the feature matrix of sampled data instances to derive a projection matrix. By fusing this matrix into the transformer's weight matrix, we obtain a compressed model. The model is subsequently fine-tuned using the entire dataset. TCSP is applied to both $T5_{base}$ and $BERT_{base}$ and evaluated on GLUE and SQuAD datasets. Remarkably, TCSP achieves a compression ratio of 44% while incurring only a 1.6% decrease in accuracy. As TCSP primarily focuses on compressing the hidden size, it can be easily combined with other methods that compress the number of filters and the attention head size, making it highly compatible.

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
