# OpenReview forum: "Transformer Compression via Subspace Projection"
_NeurIPS.cc/2023/Conference — Submitted to NeurIPS 2023_

### Official Review · Reviewer_f6zD · 2023-06-19

**Soundness:** 2 fair
**Presentation:** 3 good
**Contribution:** 2 fair
**Rating:** 3
**Confidence:** 4

**Summary:**

The paper proposes a subspace projection algorithm spanned by the features' principal components.
The projection's fusion with the different network layers is presented.
A gradient-free pruning approach is further suggested based on the parameters and activation statistics.
Finally, the proposed framework is experimented on BERT and T5 and achieves a compression ratio of 44% with at most 10 1.6% degradation.

**Strengths:**

The paper is fairly written.
The apparent main novelties of the paper are the low-rank approximation of the features and a statically based pruning approach.

**Weaknesses:**

Low-Rank approximation:\
The main weaknesses seem to arise from the comparison to prior/competing works.
For example, low-rank approximations of features have already been presented see for example
https://cs.nju.edu.cn/wujx/paper/AAAI2023_AFM.pdf \
Also, I am unsure why lines 37-38 are true: performing PCA (/Kosambi–Karhunen–Loève) is a pretty old technique for model acceleration, the subspace being defined by the principal components of the parameters or activations, this is a low-rank approximation.\
Thus, the low-rank approximation contribution of the paper should be narrowed to the definition of the data matrix.

Pruning: \
it's unclear how these simple statistics perform compared to other pruning methods or heuristics.

Experiments: \
the subspace dimension as well as the compression ratio are not given which leaves the speed-up metric subjective.
The method performance on mid-size LLM is not very good compared to old methods. \
Random projection as ablation is a very weak baseline.

Clarity:\
The paper can be refined in terms of clarity (also typos (e.g., lines 235, 282))

**Questions:**

1) The novelty issues described in the Weaknesses section should be discussed/addressed.

2) Comparison with recent techniques such as LORA and more compression metrics should be presented.

3) An ablation with SVD(weights) should be the very least comparison.

4) An ablation of the pruning should be provided

**Limitations:**

No limitations discussed

---

> ### Author Rebuttal · Authors · 2023-08-08
>
> 1. The prior work [1] and [2] are both data-aware low-rank factorization. However, unlike our approach which directly reduces the input and output dimension of the weight matrix, they still target weight matrix decomposition.  This distinnction in approach is the key  difference,  but not the low-rank factorization techniques that are employed by both their methods and ours.
> Compared to such weight matrix decomposition methods, our method has two main advantages. First, it can reduce the model's runtime memory overhead, as the dimensions of activation (i.e., both the input and the output) are reduced.  Second, weight matrix decomposition methods break down a matrix into two matrices, leading to an  increase in the total number of matrix multiplications during the inference process. This subsequently increases the overall time overhead of initiating the cuda kernel, as well as the communication overhead between the global memory and shared memory in the GPU. On the contrary, our method direclty reduces the dimension of weight matrix, without augmenting the number of matrix operations. As a result,  it avoids theseese additional overheads.
>
> 2. LoRA is an efficient fine-tuning method rather than a model compression method. This makes it fundamentally differnet from our purpose.
>
> 3. In the experimental section, we compare our method with [1] and [3]. Both of them represent improved algorithms for SVD. These papers also demonstrate that direct SVD decomposition of the weights leads to very poor model performance; therefore, we do not make a comparision with SVD applied directly to the weights.
>
> 4. In Sectino 5.2 (Table 1), w TCSP{25\%, 0\%} shows the effect without pruning , and w TCSP{25\%, 25\%} shows the effect with pruning. These exactly present the ablation studies about pruning. Meanwhile, the recent paper [4] supports the validity of our  statistics-based pruning method.
>
> 5. *the subspace dimension as well as the compression ratio are not given*
> In table 1, the notation TCSP(25%, 25%) refers to the method applied. The first 25% means the compression ratio for hidden dimension. Then for BERTbase with a hidden dimension of 768, the subspace dimension is 768*(1-25%) = 576.
>
> 6. *Random projection as ablation is a very weak baseline*
> The experiment involving random projection does not act as a baseline, but is designed to demonstrate that SVD decomposition is necessary. It shows that the success of the method is not solely due to the subsequent model fine-tuning.
>
>
> [1] Chen P, Yu H F, Dhillon I, et al. Drone: Data-aware low-rank compression for large nlp models[J]. Advances in neural information processing systems, 2021, 34: 29321-29334.
>
> [2] Yu Hao, Wu Jianxin. Compressing Transformers: Features are Low-Rank, but Weights Are Not! AAAI 2023.
>
> [3] Hsu Y C, Hua T, Chang S, et al. Language model compression with weighted low-rank factorization[J]. arXiv preprint arXiv:2207.00112, 2022.
>
> [4] Sun M, Liu Z, Bair A, et al. A Simple and Effective Pruning Approach for Large Language Models[J]. arXiv preprint arXiv:2306.11695, 2023.

---

> > ### Comment · Reviewer_f6zD · 2023-08-10
> >
> > Thank you for your answer. I think there are some misunderstandings with my review I would like to point out and elucidate.
> >
> > 1) Please correct me if I am wrong. Your main claim is the *recursive* application/approximation of $Wx \approx W(PP^{T}x)=\hat{W}\hat{x}$. In regular low-rank approximation $W\approx QQ^{T}$ you will have similar low-rank computations as the former.  The reference was to prove that data-driven low-rank approximation has been done. So as in the former review, should I understand your main claim resides in the fused data-driven subspace?
> >
> > 2) As you mentioned, Lora is an "efficient" and "fine-tuning" method (like yours). In this field, the relevance and impact are more related to the implementation and the performance rather than the exact setting. The implementation of the method on modern LLM is important but if not (as this is the case here) there is a need to at least compare with SOTA acceleration methods.
> >
> > 3)  As I wrote, a fair *ablation* study with SVD (random is quite irrelevant) is required in order to assess the impact of the different compounds of the method.
> >
> > 4) By ablation I meant regarding what I wrote before *" unclear how these simple statistics perform compared to other pruning methods or heuristics."*
> >
> > 5) By compression ratio I mean numerical compression ratio (why? c.f. Appendix: *" we choose to exclude the compression of its first layer."*) Also, not sure why the same dimensions should be adopted for all layers.
> >
> > 6) Every "ablation" is to be performed according to some baseline. The random projection was the baseline you chose which is too weak for a fair understanding and comparison (even if it proves fine-tuning is not the key to the performance. SVD of the weights, even if suboptimal would have been better).

---

> > > ### Author Response · Authors · 2023-08-11
> > >
> > > Thank you for your reply.
> > >
> > > 1. Compared to the previous work, we focus on both matrix decomposition and matrix fusing, but the previous work stays at decomposing the matrix, and they change the original matrix operation $ W * x $ to $ W_1 * W_2 * x $, while our method tries to integrate the matrix obtained from the decomposition with the original weight matrix, and change the original matrix operation $ W*x $ to $(P_0^T * W * P_1) * x$. At the same time, $(P_0^T * W * P_1)$ can be merged into a single matrix before model deployment, which has smaller input-output dimensions compared to the original matrix $W$.
> > >
> > > 2. In Table 2 and 6, we present a comparison between our method with the existing compression methods, all of which focus on model compression. These techniques are more closely related to our method than Lora is.
> > >
> > > 3. Table 3 precisely show the ablation study with SVD. We use random to replace SVD to demonstrate that the component of SVD is neccessary.
> > >
> > > 4. Since pruning is not a cirtical component, we initially overlooked the ablation of this component. However, we  could add this part of the experiment later.
> > >
> > > 5. 0.390625
> > > The total number of parameters before compression n_b is 768 * 768 *  12   12 (there are 12 layers, eacy layer contains four 768  768 matricies for MHA, and two 768  3072 for FFN, thus 768 * 768 4 + 768*3072 =768 * 768 * 12 ), and the total number of parameters after compression n_a is (ignoring the first layer) 768 * 768 * 12 + 576 * 576 * 12 * 11 + 768 * 576 * 2 (two extra matrices are used for dimensionality reduction and upscaling). Thus the compression ratio is 1 - n_a / n_b = 0.390625.
> > > The reason for using the same dimensions for all layers is because of the residuals, if we don't use the same dimensions we need to record additional matrices for the residuals for dimension changes.
> > >
> > > 6. Drone [1] has demonstrated that using SVD of the weights results in performance substantially inferior to their proposed Drone method. Thus we chose not to conduct that  redudant experiment and insteaed directly compared our approach with Drone.
> > >
> > > [1] Chen P, Yu H F, Dhillon I, et al. Drone: Data-aware low-rank compression for large nlp models[J]. Advances in neural information processing systems, 2021, 34: 29321-29334.

---

### Official Review · Reviewer_HH5t · 2023-07-02

**Soundness:** 2 fair
**Presentation:** 3 good
**Contribution:** 2 fair
**Rating:** 4
**Confidence:** 5

**Summary:**

This paper presents TCSP, a model compression approach for transformers by reducing hidden size via low-rank factorization. In addition, TCSP is compatible with other compression methods such as model pruning and head size compression. Experiment results demonstrate the effectiveness of proposed method, achieving a high compression ratio while incurring rare performance drop.

**Strengths:**

$\cdot$ This paper is well-structured and clear to understand.

$\cdot$ The algorithm is general enough, and is compatible with other compression strategies.

$\cdot$ Experiment results verify the effectiveness of the proposed method.

**Weaknesses:**

$\cdot$  The novelty of this paper is limited, the core idea resembles low-rank factorization with SVD, and the approach is more like a combination of SVD and model pruning.

$\cdot$ The author claims it is the first work to reduce the hidden size, but I doubt if the method can be successfully implemented in the industry since the lack of experimental results related to inference speed of compressed model.


**Questions:**

$\cdot$ What are the experiment settings in Tab. 2 and the ablation study?

$\cdot$ There are some writing mistakes in this paper, e.g. “fien-tune” in the header of Tab. 3.

**Limitations:**

The author has addressed the limitations and social impacts in Appendix.

---

> ### Author Rebuttal · Authors · 2023-08-08
>
> 1. Our approach is fundamentally different from existing low-rank factorization methods. While traditional methods focus on weight matrix decomposition, or method directly reduces the input and output dimension of the weight matrix.
> Compared to the weight matrix decomposition methods, our method has two main advantages. First, it can reduce the model's runtime memory overhead, as the dimensions of activation (i.e., both the input and the output) are reduced.  Second, weight matrix decomposition methods break down a matrix into two matrices, leading to an  increase in the total number of matrix multiplications during the inference process. This subsequently increases the overall time overhead of initiating the cuda kernel, as well as the communication overhead between the global memory and shared memory in the GPU. On the contrary, our method direclty reduces the dimension of weight matrix, without augmenting the number of matrix operations. As a result,  it avoids theseese additional overheads.
>
> 2. In Section 5.2 (Table 1), We explicitely report  the speed up achieved by the compression model.
>
> 3. The experimental setup of Table 2 is the same as w TCSP{25%, 25%} in Table 1.

---

> > ### Comment · Reviewer_HH5t · 2023-08-19
> >
> > For the weaknesses 2,  actually what I want to see is the speedup ratio of the compressed model on the inference engine, such as the FasterTransformer (https://github.com/NVIDIA/FasterTransformer). I have noticed the result of Tab. 1, but I didn't see the settings of hardware or inference engine.

---

### Official Review · Reviewer_Vtyh · 2023-07-05

**Soundness:** 3 good
**Presentation:** 3 good
**Contribution:** 2 fair
**Rating:** 3
**Confidence:** 3

**Summary:**

This paper proposes a decomposition-based method, called Transformer Compression via Subspace Projection (TCSP), for compressing transformers. By decomposing the feature matrix extracted by some sample data, the model is projected onto a subspace to reduce the size of hidden dimensions. Experimental results on the datasets GLUE and SQuAD show that TCSP enables 44\% parameters reduction with at most 1.6\% accuracy loss and surpassing existing methods.

**Strengths:**

This paper compresses the hidden dimension of the transformers, which is less explored.

The overall presentation of the paper is easy to understand.

**Weaknesses:**

TCSP is indeed just principal component analysis (PCA) or compressed sensing (CS), all working with the dominant subspace derived from SVD. Why another name?

I have concerns about the following aspects:

1. Motivation: From lines 51-59, this paper discusses the compression methods for transformers. Also, it mentions "We do not delve into knowledge distillation and weight sharing, as they involve training models from scratch". However, knowledge distillation (KD) includes both task-agnostic and task-specific schemes. For task-agnostic KD methods, they do not involve training models from scratch, see e.g., Wu T, Hou C, Zhao Z, et al. Weight-Inherited Distillation for Task-Agnostic BERT Compression. Meanwhile, it is a normal setting in KD to reduce the hidden size of the transformer model. Therefore, task-agnostic KD methods should be compared, too.

2. Computation: As TCSP requires SVD decomposition for a large matrix, more discussion about the computing cost and scalability is needed, especially in Table 2.

3. Robustness: Regarding the quality of subspace, how is the performance if we add noise or adversaries to the input data when generating the projection matrix? How do you ensure the sample data are representative? The timing overhead and complexity of the SVD to ensure a good projection subspace should be explicitly characterized and quantified.

Indeed, there are recent decomposition-based compression algorithms applied to transformers which the authors may benchmark against, e.g., Ren, Y., Wang, B., Shang, L., Jiang, X., \& Liu, Q. (2022). Exploring extreme parameter compression for pre-trained language models. arXiv preprint arXiv:2205.10036.



**Questions:**

See my questions above.

**Limitations:**

The models used are relatively small in size, e.g. T5-base, BERT-base. There are no experiments on the Large Language Models.

---

> ### Author Rebuttal · Authors · 2023-08-08
>
> 1. We compare our approach to the KD approaches such as DynaBERT in Appendix F.3 . We will also add the paper [1] in the comparison.
> Compared to the KD method, our method is a lightweight compression algorithm. To illustrae, while the WID method from paper [1] requires 16 hours of training on 8 A100 GPUs, our method requires only 2 hours of training on one 3090 GPU, in addition to  10 minitues for matrix decomposition. Actually, we also report this comression and training time for our method and other KD methods in Table 3. We argue that accuracy should not be the sole metric, but that the time and space efficiency should also be considered.
>
> 2. We report the time needed for SVD decomposition and model fine-tuning in Section 5.3 (Table 3). It is evident that for a BERT-base sized model, only 10 minutes are needed for matrix decomposition. Furthermore, in Appendix F.6 (Table 9), we present the results of compressing llama-7b (with a hidden size of 4096), demonstrating that our method is applicable to large-scale models. For even larger models, we can avoid performing SVD on oversized matrices by grouping the feature channels and performing the compression algorithm independently for each group.
>
> 3. We show the performance of the compression algorithm in Section 5.4 (Table 4) with different samples. The results demonstrate that good performance can be obtained by randomly picking samples on the training set. As previously discussed in answer 2, the time required for SVD decomposition is considerably less than the time needed for model fine-tuning.
>
>
> 4. *Limitations: The models used are relatively small in size, e.g. T5-base, BERT-base. There are no experiments on the Large Language Models.*
> Experiments on llama-7b are presented in Table 9 in Appendix F.6.
>
> [1] Wu T, Hou C, Zhao Z, et al. Weight-Inherited Distillation for Task-Agnostic BERT Compression[J]. arXiv preprint arXiv:2305.09098, 2023.

---

> > ### Comment · Reviewer_Vtyh · 2023-08-10
> >
> > Thanks for your feedback. Indeed, the core algorithm and almost everything thus derived is still projection onto the dominant SVD subspace (with another name & acronym). Regarding the level of innovation or novelty, I feel this falls on the low side. Regarding the benefit in a shorter runtime, it's not coupled with surprising/impressive performance either. In your rebuttal point 2, "we can avoid performing SVD on oversized matrices by grouping the feature channels and performing the compression algorithm independently for each group", why don't you do this already in the existing work and check out the tradeoff between accuracy & timing?

---

> > > ### Author Response · Authors · 2023-08-11
> > >
> > > Thank you for your reply.
> > >
> > > 1. Although both our method and traditional matrix decomposition method involve projecting into subspace, the traintional approach for each matrix operation Wx in the whole model requires performing a dimenstion reduction $ Vx $, followed by a dimension enhancement $ U(Vx) $. This results in multiple dimension reduction and enhancement throughout the whole model inference process. In contrast, thanks to the matrix fusing introduced in section 4.2, we only need to implement a dimension reduction operation at the begining of the model input (i.e.,  $ P^T $  in equation 14), and a dimension enhancement operation at the end of the model output (i.e., $ P $  in equation 14). All other matrix operations are performed in a low-dimension space within the body of the model (i.e.,  $ \hat{L}^{(1\sim N)} $ in equation 14). Thus, when compared with the traditional weight matrix decompostion method, we can reduce the matrix operations, subsequently lessening both the runtime memory overhead and cuda kernel initialization overhead.
> > >
> > > 2. We did not perform this part of the experiment because we are currently only experimenting with models of size LLMA-7b. We can still use SVD directly for matrix decomposition for models of that size. We will add this part of the experiment in the future.

---

> > > > ### Comment · Reviewer_Vtyh · 2023-08-12
> > > >
> > > > Thanks for your feedback. The technique of processing in a reduced dimensional space is commonly known as the bottleneck structure, and has been widely demonstrated in CNNs.
> > > >
> > > > A problem I noticed for this paper is the invention of new terms/acronyms for existing/known/trivial techniques to make them sound fancy.  E.g., the so-called "dimension enhancement" is simply the restoration to a higher dimension after bottleneck layer, also typical in VAEs. IMO, it is not a good practice to play with word games, just name them as they are and be down-to-earth.
> > > >
> > > > "We did not perform this part of the experiment... We can still use SVD directly ....." <-- it's exactly because the dimensions permit SVD that you can experiment with your argued "channel grouping" approach to compare its speed and accuracy vs full-size SVD. That is, it's a reason for doing, not a reason for NOT doing. In rebuttal, sometimes you're expected to furnish new supporting results to strengthen or validate your claims, not to hand-wave them away. In this rebuttal, the authors have chosen to do verbal without any new experimental results.

---

### Official Review · Reviewer_HaKw · 2023-07-05

**Soundness:** 3 good
**Presentation:** 3 good
**Contribution:** 3 good
**Rating:** 7
**Confidence:** 4

**Summary:**

This paper proposes an approach to compress the hidden size of a transformer model using subspace projection. On a high level, the paper aims at projecting the transformer model into a lower dimensional subspace using a projection matrix that is computed using a sample of the training data. This method is compared against other compression techniques using the T5 and BERT models on GLUE and SQuAD datasets and it is shown that the proposed method performs on par or better then the methods under comparison. The highlight of the experimental result is that the proposed transformer compression via subspace projection technique is able to compress models by as mush as 44% with only 1.6% degradation in performance.

**Strengths:**

The paper addresses an important problem of transformer compression. In the age of ever-increasing model sizes, it is vital to develop methods that compress large models with minimal loss in performance, if any. This paper presents a simple yet effective approach to leverage linear subspace projection for compression. The paper is easy to follow, offers sufficient literature review, and presents convincing experimental results. The experimental results are particularly strong -- 44% compression with only 1.6% loss in performance.

**Weaknesses:**

Some notation is used before definition. Could include more recent literature in Related Work section -- see below.

**Questions:**

1. What is k in line 25? Please define it before using it.

2. How are the training data sampled to estimate the projection matrix?

3. How is the rank value (k) chosen? The explanation to this and 2. are delegated to the appendix but it would be beneficial if a couple of lines addressing these are presented in main text.

4. Was the SVD solved in batch mode? Can this be done using incremental solvers instead? This may allow the use of a large matrix X.

5. What is the k value in the experimental studies? Is it 2000? Please clarify.

6. What is the average hidden layer size? What are its maximum and minimum values in the models considered?

---

> ### Author Rebuttal · Authors · 2023-08-08
>
> 1.The k in line 25 represents a dimension smaller than $d$, and in previous work, the matrix $W \in R^{d \times d}$ was decomposed into two matrices $W_1 \in R^{d \times k}$ and $W_2 \in R^{k \times d}$.
>
> 2.We randomly select samples from the training set and continue to randomly select a certain number of tokens from each sample to estimate the projection matrix.
>
> 3.The value of k is determined by the target compression rate r of the algorithm and the original hidden dimension d of the pretrained language model. It is given by the equation k = r * d.
>
> 4.We directly use the SVD solver. This solver is not suitable to very large matrices. But for large-scale models, we can group their features randomly. For each feature group, we independently calculate the projection matrix through SVD. This could approximate the accurate SVD and avoids the need to perform SVD on very large matrices.
>
> 5.As noted in answer 2, k = r * d. In the experiment, the value of k is calcualed as 768*(1-25%) = 576. The number 2000 refers to the total sampled tokens of all the sampled instances.
>
> 6.In our experiment, the minimum and maximum values of hidden layer size were 768 (BERT base) and 4096 (llama-7b), respectively.
>
> Thank you very much for your suggestions. we will add these explanations in the paper.

---

### Decision · Program_Chairs · 2023-09-21

**Decision:**

Reject

**Comment:**

This paper studied using subspace projection method to do transformer compression. The reviews of this paper has a wide range, with one accept and others on the rejection side. As I read through the reviews, one major recurring comment is the lack of novelty of this paper, its connections/differences to other low rank approximation methods for transformers. And during the rebuttal, the authors didn't provide enough evidence to convince the reviewers otherwise. Thus, I would propose the recommendation of rejection. I encourage authors also to take into account other comments to improve their paper.